# Immune-Omics Networks of *CD27*, *PD1*, and *PDL1* in Non-Small Cell Lung Cancer

**DOI:** 10.3390/cancers13174296

**Published:** 2021-08-26

**Authors:** Qing Ye, Salvi Singh, Peter R. Qian, Nancy Lan Guo

**Affiliations:** 1West Virginia University Cancer Institute, West Virginia University, Morgantown, WV 26506, USA; qiye@mix.wvu.edu (Q.Y.); ss0083@mix.wvu.edu (S.S.); peterqian3@gmail.com (P.R.Q.); 2Lane Department of Computer Science and Electrical Engineering, West Virginia University, Morgantown, WV 26506, USA; 3Department of Occupational and Environmental Health Sciences, School of Public Health, West Virginia University, Morgantown, WV 26506, USA

**Keywords:** non-small cell lung cancer (NSCLC), immunotherapy, chemotherapy, radiotherapy, biomarkers, DNA copy number variation, Boolean implication networks, CRISPR-Cas9, RNA interference (RNAi), repurposing drugs

## Abstract

**Simple Summary:**

There are currently no effective biomarkers to select chemotherapy, immunotherapy, and radiotherapy for treating lung cancer patients. This study identified genetic networks containing major immune-checkpoint inhibitors *CD27*, *PD1*, and *PDL1*, and their associated prognostic genes and proliferation genes in lung cancer tumors. A 5-gene prognostic model was developed and validated in extensive cohorts to select patients at a high risk for developing metastasis. CRISPR-Cas9 and RNA interference screening data were used in the selection of proliferation genes. These genes were associated with chemoresponse and radiotherapy response in lung cancer cell lines and patient tumors. This immune-omics network led to the discovery of repositioning drugs for improving lung cancer treatment.

**Abstract:**

To date, there are no prognostic/predictive biomarkers to select chemotherapy, immunotherapy, and radiotherapy in individual non-small cell lung cancer (NSCLC) patients. Major immune-checkpoint inhibitors (ICIs) have more DNA copy number variations (CNV) than mutations in The Cancer Genome Atlas (TCGA) NSCLC tumors. Nevertheless, CNV-mediated dysregulated gene expression in NSCLC is not well understood. Integrated CNV and transcriptional profiles in NSCLC tumors (*n* = 371) were analyzed using Boolean implication networks for the identification of a multi-omics *CD27*, *PD1*, and *PDL1* network, containing novel prognostic genes and proliferation genes. A 5-gene (*EIF2AK3*, *F2RL3*, *FOSL1*, *SLC25A26*, and *SPP1)* prognostic model was developed and validated for patient stratification (*p* < 0.02, Kaplan–Meier analyses) in NSCLC tumors (*n* = 1163). A total of 13 genes (*COPA*, *CSE1L*, *EIF2B3*, *LSM3*, *MCM5*, *PMPCB*, *POLR1B*, *POLR2F*, *PSMC3*, *PSMD11*, *RPL32*, *RPS18*, and *SNRPE*) had a significant impact on proliferation in 100% of the NSCLC cell lines in both CRISPR-Cas9 (*n* = 78) and RNA interference (RNAi) assays (*n* = 92). Multiple identified genes were associated with chemoresponse and radiotherapy response in NSCLC cell lines (*n* = 117) and patient tumors (*n* = 966). Repurposing drugs were discovered based on this immune-omics network to improve NSCLC treatment.

## 1. Introduction

Lung cancer is the leading cause of cancer death world-wide. According to the National Cancer Institute [1], 56% of lung cancer cases are at the distant stage, meaning that cancer has metastasized at the time of diagnosis. Non-small cell lung cancer (NSCLC) accounts for 84% of lung cancer cases and almost 80% of lung cancer deaths [2]. Major histology of NSCLC includes lung adenocarcinoma (LUAD), squamous cell lung carcinoma (LUSC), and large cell carcinoma. The major treatment for early-stage NSCLC is surgical resection. Nevertheless, stage I NSCLC patients have a recurrence/metastasis rate of about 22–38% within five years post-surgery [3]. It remains a challenge to select early-stage NSCLC patients for more aggressive treatment. NSCLC patients with stage II and above receive chemotherapy, with additional radiation for stage III and IV patients [4]. Although adjuvant chemotherapy of stage II and stage III disease has resulted in 10–15% increased overall survival [5], the prognosis for early-stage NSCLC remains poor [6]. These data indicate that many patients may not have benefited from the prescribed chemotherapy. To date, there are no biomarkers to predict the risk for recurrence/metastasis and the clinical benefits of specific chemotherapy in resectable NSCLC of all histological subtypes in clinics.

Immunotherapy has rapidly emerged as an effective and less toxic treatment than chemotherapy in patients with advanced lung cancers [7,8,9,10]. A paired single-cell analysis comparing normal lung tissue and blood with tumor tissue in stage I NSCLC found that early-stage tumors had already begun to alter the immune cells in their microenvironment [10]. These results provided evidence that immunotherapy has potential use to treat early-stage lung cancer patients. More recently, studies showed that immune-checkpoint inhibitors (ICIs) in combination with chemotherapy could improve NSCLC patient survival regardless of PDL1 expression [11]. PDL1 and tumor mutational burdens are not proven indicative biomarkers. To date, predictive biomarkers of immunotherapy are not well established except PD1 or PDL1, and it is unlikely that a single marker is sufficient.

Previously we identified a 7-gene NSCLC prognostic and chemo-predictive qRT-PCR assay, including *CD27* [12]. The protein expression of CD27 quantified with ELISA had a strong correlation with its mRNA expression in NSCLC tumors [12]. CD27 is a new generation of ICI [13] and is being tested as adjuvant therapy in phase I/II clinical trials for multiple tumor types with promising results [14,15]. Increasing evidence supports that CD27 agonist antibodies, either alone or combined with PD1-blockade, can improve the therapeutic efficacy of cancer vaccines and immunotherapy in general [16,17]. Anti-CD27/CD20 showed therapeutic potential in recruiting and activating myeloid cells for enhanced killing of tumors [18]. A review of The Cancer Genome Atlas (TCGA) data showed that *CD27*, *PD1*, *PDL1*, and *CD20* had a higher frequency of DNA copy number variations (CNV) than mutations in LUAD and LUSC tumors (Figure A1, Figure A2, Figure A3, Figure A4, Figure A5 and Figure A6). Nevertheless, CNV-mediated dysregulated gene expression in NSCLC is not well understood. FoundationOneDx assay is an FDA-approved companion test for tumor mutation burdens. Currently, there are no effective CNV or mRNA expression-based biomarkers for the selection of immunotherapy, chemotherapy, and radiotherapy in individual NSCLC patients.

The tumor immune microenvironment is a complex network of genes and proteins functioning in immune and stromal cells as well as other systemic host factors, which together determine the immunological status of the host–tumor interactions [19]. Given this multidimensionality, a gene signature that integrates these elements should be developed to select patients for personalized immunotherapy [19], in combination with chemotherapy and/or radiotherapy. Recent studies systematically revealed genomic instability and transcriptomic dysregulation in tumor immune microenvironment [20,21]. Innovative network approaches are needed to identify interweaving molecular interactions with implications in prognosis, proliferation, and response to chemotherapy, immunotherapy, and radiotherapy for improving cancer treatment. Such pathway and network approaches will lead to the discovery of novel therapeutic targets as well as repositioning drug candidates [22]. Here, we developed a novel Boolean implication network scheme to model CNV and gene expression (GE) profiles in NSCLC tumors (*n* = 371) to decipher molecular networks for *CD27*, and dissect those involving *CD27*, *PD1*, and *PDL1*. Such revelation will provide insights into molecular mechanisms underlying immunotherapy targeting CD27 alone, or in combination with the PD1/PDL1-blockade. Significantly enriched cytobands, pathways, and gene families were analyzed for the constructed *CD27* networks. Based on the identified multi-omics network of *CD27*, *PD1*, and *PDL1*, proliferation genes were identified from CRISPR-Cas9 (*n* = 78) and RNAi (*n* = 92) screening data; prognostic genes were selected and validated in NSCLC tumors (*n* = 1163). The identified genes were examined for their association with chemotherapy in the Cancer Cell Line Encyclopedia (CCLE) NSCLC panel (*n* = 117) and radiotherapy in TCGA patient tumors (*n* = 966). The immune infiltration associated with the identified genes was investigated in myeloid dendritic cells, macrophages, neutrophils, CD4+ T cells, CD8+ T cells, and B cells. Finally, repurposing drug candidates for NSCLC treatment were discovered based on the identified immune-omics network. The overall study scheme is shown in Figure 1.

## 2. Materials and Methods

### 2.1. Patient Cohorts

#### 2.1.1. NSCLC Patient Cohort GSE31800

DNA copy number profiles of 271 NSCLC tumor samples were measured in a previous study [23], including 179 adenocarcinomas and 92 squamous cell carcinomas (raw data available at NCBI GEO with the accession number GSE31800). The DNA copy number profiles were measured with whole genome tiling path array CGH. A total of 49 out of the 271 samples had matched custom microarray gene expression profiles [23], with 29 adenocarcinomas and 20 squamous cell carcinomas.

#### 2.1.2. NSCLC Patient Cohort GSE28582

A publicly available NSCLC patient cohort (*n* = 100) [24,25] was included in this study, including 50 adenocarcinomas, 22 large cell carcinomas, and 28 squamous cell carcinomas (raw data available at NCBI GEO with the accession number GSE28582). All the samples had matched DNA copy number variation profiled by SNP array and microarray gene expression data. Patients who survived shorter than 20 months after treatment were defined as short-term survivors, and those who survived longer than 58 months after treatment were defined as long-term survivors.

#### 2.1.3. NSCLC Patient Cohort GSE81089

A total of 199 NSCLC patient tumors were collected in a previous study [26]. Deep RNA sequencing of the patient tumors was generated with Illumina HiSeq 2500 (raw data available at NCBI GEO with the accession number GSE81089). Patients with sufficient survival information (*n* = 197) were included in this study.

#### 2.1.4. TCGA NSCLC Patient Cohorts

The TCGA-LUAD (*n* = 515) and TCGA-LUSC (*n* = 501) data were available from LinkedOmics [27] (http://linkedomics.org/, accessed on 28 April 2021). The gene expression was quantified with Illumina HiSeq 2000 RNA Sequencing platform (Illumina, San Diego, CA, USA). The RNA-Seq data values were log-transformed. The TCGA-LUSC and TCGA-LUAD dataset also had the cancer stage and radiotherapy information for each sample.

### 2.2. Data Pre-Processing

#### 2.2.1. CNV Data Pre-Processing

The original copy number data of GSE31800 was converted from human genome assembly Hg18 to Hg38 in this study. According to the converted chromosomal locations, 26,061 genes were matched. Bioconductor R package “CGHbase” (v1.46.0) [28] and “CGHcall” (v2.48.0) [29] were used to call the gains and losses of the copy number data.

The original GSE28582 data [24,25] were processed by converting the Hg17 reference to Hg38 in this study. According to the converted chromosomal locations, 12,842 genes were matched. PennCNV package [30] was used to process the SNP array data. The CNV data were categorized into 3 levels: 1 (amplification), -1 (deletion), and 0 (normal).

#### 2.2.2. Gene Expression Data Pre-Processing

To define the level of gene expression, 27 housekeeping genes (ACTB, B2M, CDKN1B, ESD, FLOT2, GAPDH, GRB2, GUSB, HMBS, HPRT1, HSP90AB1, IPO8, LDHA, NONO, PGK1, POLR2A, PPIA, PPIH, PPP1CA, RHOA, RPL13A, SDCBP, TBP, TFRC, UBC, YAP1, YWHAZ) [12,31,32,33,34] were used to define the thresholds of gene expression categorization. The total percentage of over-expression and under-expression samples for all the housekeeping genes was fixed to be 30%, and the number n which meets the following Equation (1) was computed for each gene [34,35]:(1)$$\begin{array}{c} \left(the\;number\;of\;samples\;with\;expression>mean+n\times std\right)\\ +\left(the\;number\;of\;samples\;with\;expression<mean-n\times std\right)\\ =30\%\;of\;the\;total\;number\;of\;samples \end{array}$$

The averaged *n* value of all 27 housekeeping genes was 0.874 in the dataset GSE31800 and 0.977 in the dataset GSE28582. These *n* values were used to categorize each gene expression dataset. The mean and standard deviation for each gene were calculated. The sample with a gene expression higher than (mean + *n* × standard deviation) was defined as over-expressed; the sample with a gene expression lower than (mean − *n* × standard deviation) was defined as under-expressed; the sample with a gene expression in between the two thresholds was defined as normal. After categorizing with the above-mentioned thresholds, more than 50% of the genes in both patient cohorts had more over-expressed or under-expressed samples than the housekeeping genes.

### 2.3. Boolean Implication Networks

The implication network algorithm used in this study was proposed by Guo et al. [36,37], based on prediction logic [38]. Formal logic rules relating two binary variables were inferenced for each pair of genes to form the implication network. In this study, the algorithm was improved by counting the error cell of each implication rule such that it can model not only the binary variable but also multivariate data. This network induction algorithm is generic for various general applications, including multinomial datasets and multi-classification problems.

The Boolean implication algorithm evaluates six implication rules: A → B (positive implication), A → ¬ B (forward negative implication), ¬ A → B (inverse negative implication), ¬ A → ¬ B (negative implication), A ⟷ B (positive equivalence), and A ⟷ ¬ B (negative equivalence). (A & ¬ B) is the error cell for A → B. (A & B) is the error cell for A → ¬ B. (¬ A & ¬ B) is the error cell for ¬ A → B. (¬ A & B) is the error cell for ¬ A → ¬ B. (A & ¬ B) and (¬ A & B) are the error cells for A ⟷ B. (A & B) and (¬ A & ¬ B) are the error cells for A ⟷ ¬ B.

The Boolean implication algorithm relies predominantly upon error cells for a particular rule while calculating the scope (U_p_), indicating how much the data is covered, and precision (∇_p_), indicating accuracy, for that rule. For instance, while looking for co-amplification in a pair of genes (A→B), the algorithm checks if a gene is amplified, and the remaining two states of deletion and normalcy become simply the negate of amplification; the error would be when gene A is amplified but gene B is not amplified. In this manner, the algorithm can use the same fundamental principle to create implication relations amongst genes, irrespective of the number of states for each gene.

For each implication rule to be significant, it must have a scope and precision value which is greater than the threshold value, and the greatest amongst all the rules for the pair of variables in question. The threshold for scope and precision depends on the significance level in statistical tests. The Algorithm 1 was described as:
**Algorithm 1**Begin  Set a significance level ∇_min_ and a minimal U_min_  For node_i_, i ∈ [0, ν_max_ − 1] and node_j_, j ∈ [i + 1, ν_max_]  (Note: ν_max_ is the total number of nodes)     For all empirical case samples *N*, compute a contingency table:$$M_{\mathrm{ij}}=\begin{array}{cc} N_{11} & N_{12}\\ N_{21} & N_{22} \end{array}$$     For each relation type *k* out of the six cases find the solution:Max U_p_Subject to     Max U_p_ > U_min_∇_p_ > ∇_min_∇_error cells_ > ∇_non-error cells_     If the solution exists, then return a type *k* relationEnd

In the contingency table Mij, i stands for the index of the first gene (A) in the methodology, and j stands for the index of the second gene (B). We tested one association type (i.e., co-amplification, etc.) each time. N_11_ represents the number of samples where both A and B have a true statement in the examined association type; N_12_ is the count of samples where A is true, but B is not true; N_21_ is the count of samples where A is not true, but B is true; N_22_ is the number of samples where both A and B are not true. For a single error cell, where N_ij_ is the number of error occurrences, N_i._ is the row sum of contingency table M_ij_, and N_.j_ is the column sum of contingency table M_ij_. N is the total number, which is the sum of N_11_, N_12_, N_21_, and N_22_. The scope U_p_ and the precision ∇_p_ is defined as (2):(2)$$U_{p}=U_{ij}=\frac{N_{i.}\times N_{.j}}{N^{2}},\;\nabla_{p}=\nabla_{ij}=1-\frac{N_{ij}}{N\times U_{p}}$$

For multiple error cell types, the scope U_p_ and the precision ∇_p_ is defined as the Equation (3), where ω_ij_ = 1 for error cells; otherwise, ω_ij_ = 0.
(3)$$U_{p}=\sum_{i}\sum_{j}\omega_{ij}\times U_{ij},\;\nabla_{p}=\sum_{i}\sum_{j}\left(\frac{\omega_{ij}\times U_{ij}}{U_{p}}\right)\nabla_{ij}$$

The pairwise gene relationships at the CNV level included co-amplification, co-deletion, amplification-deletion, and deletion-amplification. The pairwise gene relationships at gene expression level included co-upregulation, co-downregulation, upregulation-downregulation, and downregulation–upregulation. The modelled CNV and gene expression cross-level included amplification–upregulation, amplification–downregulation, deletion–upregulation, and deletion–downregulation. We can adjust the scope and precision threshold by altering the statistical significance for the implication rules in our implementation. The threshold value was calculated from a one-tailed *z*-test based on the sample size by setting the ideal *z* value. The *z* value used for this study was 1.64 (95% confidence interval, α = 0.05, one-tailed *z*-tests).

### 2.4. Functional Enrichment Analysis Using ToppGene

ToppFun of the ToppGene suite [39] was used to detect functional enrichment including pathways, gene families, and cytobands. The ToppFun tool used the false discovery rate (FDR) multiple correction method with a significance cutoff level of 0.05 in the enrichment analysis.

### 2.5. Ingenuity Pathways Analysis

Functional involvement of genes in cancer, immunological disease, and inflammatory disease was examined with Qiagen Ingenuity Pathways Analysis (IPA) [40] (Ingenuity^®^ Systems, Qiagen, Hilden, Germany). IPA is a functional pathway analysis tool incorporating genes, cellular species such as proteins, and chemical compounds with information curated from the published literature.

### 2.6. Cancer Cell Line Encyclopedia (CCLE)

Gene expression data for CCLE were downloaded from DepMap 20Q2 (https://figshare.com/articles/dataset/DepMap_20Q2_Public/12280541, accessed on 1 April 2021) [41]. Gene expression data were obtained from the CCLE data portal (https://data.broadinstitute.org/ccle/CCLE_RNAseq_081117.rpkm.gct, accessed on 1 April 2021). RNA-seq data were quantified using the GTEx pipelines [42]. A total of 117 NSCLC cell lines were included in this analysis.

### 2.7. CRISPR-Cas9 Assays

Gene knockout effects in CCLE using CRISPR-Cas9 screens were quantified in Project Achilles [43,44]. The data were obtained from DepMap 20Q2 (https://figshare.com/articles/dataset/DepMap_20Q2_Public/12280541, accessed on 1 April 2021) [41]. The CRISPR-Cas9 data were processed with the CERES method [43]. Gene effects in each cell line were normalized such that the median non-essential gene knockout effect is 0 and the median essential gene knockout effect is −1. A gene is defined as an essential gene if it is essential to the cell growth in each line; otherwise, it is defined as a non-essential gene. There were 78 NSCLC cell lines with genome-scale CRISPR-Cas9 knockout results. A normalized dependence score less than −0.5 indicates a significant effect in CRISPR-Cas9 knockout.

### 2.8. RNAi Functional Assays

Genome-scale RNAi screening data in CCLE were obtained from Project Achilles (https://depmap.org/R2-D2/, accessed on 1 April 2021) [45]. The DEMETER2 method [45] was used to estimate average gene dependency scores in each cell line for short hairpin RNA (shRNA) libraries. Gene dependency scores were standardized with DEMETER2 such that the median of the across-cell-line average dependency scores of the positive control gene set was -1 and that of the negative control gene set was 0. There were 92 NSCLC cell lines with genome-scale RNAi screening results normalized with DEMTER2. A normalized dependence score less than −0.5 indicates a significant effect in RNAi knockdown.

### 2.9. Immune Infiltration Estimation

TIMER 2.0 [46,47,48] was used to find the association of gene expression and immune infiltration. TIMER is a comprehensive resource for systematically analyzing the immune infiltration of different cancer types (http://timer.cistrome.org/, accessed on 14 June 2021). It provides the abundance of immune infiltration estimated by a variety of immune deconvolution methods.

### 2.10. PRISM Drug Response in CCLE

The growth inhibitory activity of 4,518 drugs was quantified in 578 human cancer cell lines using the PRISM molecular barcoding and multiplexed screening method [49]. The PRISM dataset is available at the Cancer Dependency Map portal (https://depmap.org/portal/download/, accessed on 1 April 2021). Drug responses of 9 commonly used chemotherapeutic regimens in treating NSCLC were included in this study: carboplatin, cisplatin, paclitaxel, docetaxel, gemcitabine, vinorelbine, etoposide, gefitinib, and erlotinib. For each drug, cell lines with IC_50_ or EC_50_ value higher than the maximum dose were defined as resistant; cell lines with IC_50_ or EC_50_ value lower than the minimum dose were defined as sensitive. The remaining cell lines were divided into groups of resistant, sensitive, or partial response by using the mean ± 0.5 standard deviation (SD) of the drug activity metrics, including IC_50_, ln(IC_50_), EC_50_, or ln(EC_50_) values [50,51]. Cell lines with a drug activity value less than the mean − 0.5 SD were defined as sensitive to the drug, and those with a drug activity value between the mean + 0.5 SD and the mean − 0.5 SD were defined as having a partial response to the drug. Cell lines with a drug activity value greater than the mean + 0.5 SD were defined as resistant to the drug. This categorization corresponds to the RECIST 1.1 system (i.e., complete response, partial response, and stable disease/disease progression) in evaluating chemotherapeutic response in solid tumors [52].

### 2.11. Genomics of Drug Sensitivity in Cancer (GDSC1/2)

Drug screening data were downloaded from Genomics of Drug Sensitivity in Cancer (GDSC) Project (https://www.cancerrxgene.org/downloads/bulk_download, accessed on 15 April 2021) [53]. The GDSC Project screened more than 1,000 genetically characterized human cancer cell lines with a wide range of anti-cancer therapeutic agents. Among the commonly used chemotherapeutic regimens in treating NSCLC drugs, nine were found in GDSC1, including cisplatin, docetaxel, erlotinib, etoposide, gefitinib, gemcitabine, pemetrexed, and vinorelbine; seven were found in GDSC2, including cisplatin, docetaxel, erlotinib, gefitinib, gemcitabine, paclitaxel, and vinorelbine. For each drug, cell lines were defined as resistant, sensitive, or partial response by using the mean ± 0.5 SD of the IC_50_ values as described above.

### 2.12. Drug Repurposing Using Connecitivity Map (CMap)

CMap (https://portals.broadinstitute.org/cmap/, accessed on 7 April 2021) [54] was used to identify candidate small molecules as repurposing drugs based on our gene expression signatures. CMap is a database of gene expression profiles of pharmacologically perturbed cell lines. It allows the use of gene expression signatures to connect small molecules, genes, and disease.

### 2.13. Statistical Methods

Statistical analysis was performed using Rstudio version 1.4.1106 [55]. Differential gene expression between two groups was evaluated with Student’s *t*-tests, and a two-sided *p*-value < 0.05 was considered statistically significant. Survival analysis was performed using Kaplan–Meier analysis with the survival package in R. Log-rank tests were used to assess the difference in survival probability from different groups in Kaplan–Meier analyses. The genes in the networks were evaluated with the Cox proportional hazards model and Kaplan–Meier analysis. Pearson’s correlation test was used to find the relationship between two variables. The multivariate Cox regression analysis was used to build the model of risk scores.

## 3. Results

### 3.1. Construction of Multi-Omics CD27 Networks

CD27 networks containing co-occurrences of DNA copy number aberrations, co-expression, and CNV-mediated gene expression dysregulations were generated with two NSCLC patient cohorts (GSE31800 and GSE28582) using the Boolean implication network algorithm. The significant genetic associations (*p* < 0.05, *z*-tests) found in both patient cohorts were kept for further analysis. There were 111 genes associated with CD27 from the CNV level, 75 genes from the gene expression level, and 24 genes from the CNV-mediated gene expression (GE) level (Figure 2). The detailed information of each associated gene pair was provided in Appendix A.

The enrichment of the *CD27* networks in cytobands, gene families, and pathways was analyzed with ToppGene. The perturbed *CD27* CNV network was overrepresented (*p* < 0.05, FDR < 0.05) in cytobands 12p11, 12p12, and 12p13 (Figure 2A). These genes were significantly (*p* < 0.05, FDR < 0.05) enriched in gene families, including proline-rich proteins, G protein-coupled receptors, class C orphans, complement system, potassium voltage-gated channels, apolipoproteins, and calcium voltage-gated channel subunits. The enriched (*p* < 0.05, FDR < 0.05) pathway was oxidative damage. Interestingly, *LAG3*, a promising ICI [13], had a significant (*p* < 0.05, *z*-tests) co-amplification with *CD27* in both patient cohorts GSE31800 and GSE28582. The computationally detected patterns showed when *CD27* was amplified in the NSCLC tumors, *LAG3* was also amplified; when CD27 was not amplified in the NSCLC tumors, *LAG3* was not amplified either (Appendix A). Together with *LAG3*, numerous genes on cytoband 12p13 had a significant (*p* < 0.05, *z*-tests) CNV co-occurrence with *CD27* (Figure 2A, Appendix A).

The perturbed *CD27* gene expression (GE) network (Figure 2B) was significantly (*p* < 0.05, FDR < 0.05) enriched in gene families including CD molecules/tumor necrosis factor superfamily, purinergic receptors P2X/deafness-associated genes, receptor tyrosine kinases/CD molecules/immunoglobulin-like domain, apolipoproteins, butyrophilins, chemokine ligands/endogenous ligands, and formyl peptide receptors. The enriched pathways (*p* < 0.05, FDR < 0.05) were cancer immunotherapy by PD-1 blockade, IL12-mediated signaling events, downstream signaling in naïve CD8+ T cells, natural killer cell-mediated cytotoxicity, type II interferon signaling (IFNG), interactions between immune cells and microRNAs in tumor microenvironment, granzyme A-mediated apoptosis pathway, and calcineurin-regulated NFAT-dependent transcription in lymphocytes. These genes were overrepresented (*p* < 0.05, FDR < 0.05) in cytobands 1p22.2, 6p25-p23, 9q32-q34.11, and 9q22.32-q31.3 (Figure 2B, Appendix A).

The *CD27* CNV-mediated gene expression (GE) network (Figure 2C) was significantly (*p* < 0.05, FDR < 0.05) enriched in gene families including adiponectin receptors, COP9 signalosome, progestin and adipoQ receptor family, peroxins, poly(ADP-ribose) polymerases, and INO80 complex/DNA helicases. These genes were overrepresented (*p* < 0.05, FDR < 0.05) in cytobands 2p21-p22, 4q21, 12p11-p13, and 19q13 (Figure 2C). No significantly overrepresented pathways were found in the *CD27* CNV-mediated GE network. Details were provided in Appendix A.

Multiple genes in the identified *CD27* networks had experimentally confirmed interactions with CD27 as retrieved with IPA (Figure 3). In cancer, immunological disease, and inflammatory disease, CD27 activates NF-κB [56,57,58]. Cytokines IL-2 and IL-15 downregulate CD27 [59]. The binding [60,61,62,63] and interaction [64,65,66,67,68,69,70,71,72,73,74,75,76,77,78,79] of CD27 with CD70 make CD27-CD70 an attractive therapeutic target [80]. In the Boolean implication networks generated in this study, a downregulation of *FCRL3* was associated with a downregulation of *CD27* and *CD20* (*MS4A1*), respectively, in both patient cohorts GSE31800 and GSE28582 (results not shown). *FCRL3* may be involved in human-specific mechanisms to control the generation of natural T regulatory (nTreg) cells, and its expression on nTreg cells correlates with cell hypoproliferation [81]. The expression of *FOXP3* alone is insufficient to induce *FCRL3* expression [81]. The co-expression associations among *CD27*, *CD20*, and *FCRL3* observed in multiple NSCLC patient cohorts in this study warrant further investigation. Detailed experimentally confirmed interactions retrieved from IPA were provided in Appendix A.

### 3.2. Delineation of CD27, PD1, and PDL1 Multi-Omics Networks

Multi-omics association networks of *CD27*, *PD1* (*PDCD1*), and *PDL1* (*CD274*) were also generated with the Boolean implication network algorithm. In the first layer, the following genes were selected. First, genes that had a significant (*p* < 0.05, *z*-tests) co-expression between *CD27* and *PD1*, or between *CD27* and *PDL1*, in both patient cohorts GSE31800 and GSE28582 were identified in the genome-scale analysis. Second, genes with a significant (*p* < 0.05, *z*-tests) co-occurrence of *CD27* and *PDL1* DNA copy number aberrations in patient cohort GSE31800 were identified, since the CNV information of *PDL1* was not available in GSE28582. Third, genes with a significant (*p* < 0.05, *z*-tests) CNV-mediated gene expression (GE) association between *CD27* and *PD1*, or between *CD27* and *PDL1*, in patient cohort GSE31800 were selected. There were 38 genes selected at the CNV level, 88 genes at the GE level, and 26 genes at the CNV-mediated GE level (a detailed gene list is provided in Appendix A). From this gene list, genes that were significantly associated with survival in GSE81089 (*p* < 0.05, univariate Cox-modeling) and were validated with a concordant significant differential expression in short versus long survival (*p* < 0.05, two sample *t*-tests) in GSE28582 were selected as prognostic genes in layer 1, including *SPP1*, *FPR1*, *TGM5*, and *FOSL1* (Figure 4A). *FOSL1* connects the *KRAS* oncogene to components of the mitotic pathway [82]. Mutant *KRAS* lung and pancreatic cancer patients with high *FOSL1* expression had the worst survival outcome [82]. *FOSL1* inhibition is detrimental to both KRAS-driven lung and pancreatic cancer [82]. MORAb-202, an anti-human FRα (FOSL1) antibody-drug conjugate achieved robust antitumor response in patient-derived xenograft models [83] and phase I study of FRα (FOSL1)-positive advanced solid tumors [84]. *SPP1* promotes proliferation, migration, invasion, and chemoresistance to cisplatin in NSCLC cells [85], and is important in mediating macrophage polarization and facilitating immune evasion in lung adenocarcinoma through the upregulation of *PDL1* [86]. In addition, genes with a significant dependency score in 50% of the tested NSCLC cell lines in CRISPR-Cas9 (*n* = 78) or RNAi (*n* = 92) were defined as layer 1 proliferation genes, including *URM1*, *VIRMA*, *BUB3*, *POLA1*, *NDUFS8*, *OR4K1*, and *MYC* (Figure 4A).

Similarly, genes with a significant association (*p* < 0.05, *z*-tests) with at least 2 out of the 14 genes (*CD27*, *PD1*, *PDL1*, and 11 layer 1 genes) in both patient cohorts GSE31800 and GSE28582 were selected as layer 2 candidate genes using the Boolean implication networks. A total of 30 genes at the CNV level, 1538 genes at the GE level, and 24 genes at the CNV-mediated GE level were selected (a detailed gene list is provided in Appendix A). From this list of the layer 2 genes, 33 genes were significantly associated with survival in the patient cohort GSE81089 *(p* < 0.05, univariate Cox-modeling) and were validated with a concordant significant differential expression in short versus long survival (*p* < 0.05, two sample *t*-tests) in the patient cohort GSE28582. When fitting these 33 genes in a multivariate Cox model in a separate NSCLC patient cohort GSE81089 (*n* = 197), five significant (*p* < 0.05, multivariate Cox modeling) genes, including *EIF2AK3*, *F2RL3*, *FOSL1*, *SLC25A26*, and *SPP1*, were selected as layer 2 prognostic genes. From the list of the layer 2 candidate genes, 13 genes (*COPA*, *CSE1L*, *EIF2B3*, *LSM3*, *MCM5*, *PMPCB*, *POLR1B*, *POLR2F*, *PSMC3*, *PSMD11*, *RPL32*, *RPS18*, and *SNRPE*) had a significant dependency score in 100% of the tested NSCLC cell lines in both CRISPR-Cas9 (*n* = 78) and RNAi assays (*n* = 92), defined as layer 2 proliferation genes.

The final immune-omics network (Figure 4A) contained genes having a direct or indirect association with *CD27*, *PD1*, and *PDL1*, with each network edge indicating a co-occurrence of CNV, co-expression, or CNV-mediated gene expression dysregulation in NSCLC tumors observed in multiple patient cohorts. This network contained both NSCLC prognostic genes and proliferation genes. Like *CD27*, *PD1*, and *PDL1*, the identified NSCLC prognostic genes and proliferation genes in the immune-omics network had a higher occurrence of CNVs than mutations in TCGA-LUAD and LUSC tumors (Table A1 and Table A2). The correlation of immune infiltration with the expression of 21 genes (*CD27*, *PD1*, *PDL1* and the layer 2 genes) in TCGA-LUAD (*n* = 515) and TCGA-LUSC (*n* = 501) patient cohorts was assessed with TIMER (Figure 4B–C) [46,47,48]. The immune infiltration included myeloid dendritic cells, macrophages, neutrophils, CD4+ T cells, CD8+ T cells, and B cells. The detailed information was provided in Appendix A.

Using the five prognostic genes selected in the layer 2 (Figure 4A), a multivariate Cox model was built in the patient cohort GSE81089 (*n* = 197) to calculate the risk-score for each patient (Figure 5). The Kaplan–Meier analysis results showed that the patients who had a risk-score less than 2.25 survived significantly (*p* = 3 × 10^−7^, HR: 3.752 (2.175, 6.474)) longer than the patients who had a risk-score equal or greater than 2.25 in the training cohort GSE81089 (Figure 5A). The mRNA expression data of TCGA-LUAD and TCGA-LUSC was used to validate this prognostic model (Figure 5B). The Kaplan–Meier analysis results in the TCGA data (*n* = 966) also showed a significantly better 10-year survival (*p* = 0.02, HR: 1.521 (1.073, 2.158)) in the patient group with lower risk-scores, confirming the results from the training data.

### 3.3. Identification of Genes Associated with Radiotherapy and Chemotherapy

According to the current practice guidelines, NSCLC patients with stage II and above receive chemotherapy, with additional radiation for stage IIIA patients [4]. To identify genes associated with radiotherapy response, stage III and IV patients in TCGA-LUAD and TCGA-LUSC who had received radiotherapy were included in the analysis. Among the genes identified in the immune-omics network (Figure 4A), *CD27*, *MYC*, and *URM1* had a significant differential expression (*p* < 0.05, two sample *t*-tests) in the short survival (<20 mo.) versus long survival (>58 mo.) patient group (Figure 5C). We recognize that many factors other than the response to radiotherapy may impact survival in this studied patient cohort. *CD27* and *URM1* expressed higher in the short survival patient group, and *MYC* expressed higher in the long survival patient group. The main function of URM1, a ubiquitin-like protein, is to modify tRNA, and elevated levels of tRNA modifying enzymes promote tumorigenesis and metastasis [87]. The *MYC* oncogene causes many human cancers [88] and is activated by *FOXP3* in NSCLC [89].

Furthermore, drug responses of 10 commonly used chemotherapeutic regimens in treating NSCLC were included in this study, including carboplatin, cisplatin, paclitaxel (Taxol), pemetrexed, docetaxel, gemcitabine, vinorelbine, etoposide, gefitinib, and erlotinib. The studied NSCLC cell lines included adenocarcinoma, squamous cell carcinoma, large cell carcinoma, and adenosquamous carcinoma. Due to the synergism and successful results of the combination of cisplatin-etoposide in treating small cell lung cancer, long-term daily administration of oral etoposide in combination with cisplatin was used to treat NSCLC [90]. A systematic review showed that cisplatin-etoposide has comparable efficacy as carboplatin-paclitaxel when used with concurrent radiotherapy for patients with stage III unresectable NSCLC [91]. Paclitaxel, a tubulin-binding agent, is commonly used to treat NSCLC in combination with a platinum-based compound [92]. Gefitinib and erlotinib are widely used epidermal growth factor receptor (EGFR) tyrosine kinase inhibitors for treating advanced NSCLC with proven efficacy. A recent meta-analysis showed that gefitinib and erlotinib have comparable effects on patient survival, overall response rate, and disease control rate, with no considerable variation associated with EGFR mutation status, ethnicity, line of treatment, and baseline brain metastasis status [93]. Docetaxel offers clinical benefits as a second-line treatment of NSCLC in patients previously treated with platinum-based chemotherapy [94]. It was recently reported that the combination of pembrolizumab (anti-PD1 immunotherapy) plus docetaxel was well tolerated and substantially improved progression-free survival and overall response rate in patients with advanced NSCLC after platinum-based chemotherapy, including patients with EGFR variations [95]. Here, genes associated with chemoresponse were selected based on their differential expression in sensitive versus resistant NSCLC cell lines to the studied drugs. In the identified *CD27*, *PD1*, and *PDL1* network (Figure 4A), multiple genes were associated with chemosensitivity or chemoresistance to cisplatin, docetaxel, erlotinib, etoposide, gefitinib, gemcitabine, paclitaxel, pemetrexed, and vinorelbine in the CCLE NSCLC cell lines (*n* = 117, Table 1).

### 3.4. Discovery of Repurposing Drug Candidates

The identified *CD27*, *PD1*, and *PDL1* network (Figure 4A) was used to discover repurposing drugs with potential implications in improving NSCLC treatment in combination with immunotherapy. The following mechanisms of action were considered in the selection of small molecules: (1) to downregulate *CD27*, *PD1*, and *PDL1* expression; (2) to inhibit proliferation genes that had a significant impact in 100% of tested NSCLC cell lines in CRISPR-Cas9 and RNAi assays; (3) to maintain a high expression of survival protection genes that expressed higher in the long survival patient group; (4) to downregulate the expression of survival hazard genes that expressed higher in the short survival patient group. The identified survival protection genes and hazard genes had consistent expression patterns in three NSCLC cohorts (GSE81089, GSE28582, and GSE81089) as well as TCGA. This gene expression signature was input into CMap to generate a list of significant compounds (Figure 6A). *PDL1* was not available in the CMap database and was not included in the input gene list in Figure 6A.

A total of 119 significant (*p* < 0.05) compounds were identified with CMap [54] (detailed information provided in Appendix A). These compounds were further selected by using the following criteria in the CCLE NSCLC panel (*n* = 117): (1) choosing the small molecules with strong inhibitory effects of NSCLC cells, i.e., with low IC_50_ and EC_50_ values, and (2) filtering the compounds with a negative correlation of drug concentration and mRNA expression of *CD27*, *PD1*, or *PDL1* in the NSCLC cell lines. In the original PRISM drug screen, eight doses ranging from 0.0006 μM to 10 μM were tested on most of the compounds [49]. Four small molecules (anisomycin, disulfiram, doxorubicin, and mitoxantrone) had a small average IC_50_ or EC_50_ value (<1 μM) in the PRISM drug screen and were selected for drug repurposing (Figure 6B), since most NSCLC cell lines were relatively sensitive to these small molecules. Seven compounds (puromycin, aztreonam, ivermectin, piperlongumine, trichostatin A, irinotecan, and mitoxantrone) had a significant negative correlation (*p* < 0.05, Pearson’s correlation) of drug concentration (IC_50_, EC_50_, ln(IC_50_), or ln(EC_50_) value) with mRNA expression of *CD27*, *PD1*, or *PDL1*, respectively, in the datasets PRISM, GDSC1, or GDSC2 (Figure 7).

Doxorubicin, mitoxantrone, and irinotecan are chemotherapeutic drugs. Doxorubicin is used to treat breast cancer, bladder cancer, Kaposi’s sarcoma, lymphoma, and acute lymphocytic leukemia. TIGAR knockdown may inhibit epithelial-to-mesenchymal (EMT) transition in doxorubicin-resistant human NSCLC [96]. Mitoxantrone is mainly used to treat acute myeloid leukemia, with reported excellent in vitro antitumor activity against human lung adenocarcinoma [97]. Irinotecan is used for the treatment of colon cancer and small cell lung cancer. A randomized phase III clinical trial showed that cisplatin plus irinotecan, carboplatin plus paclitaxel, cisplatin plus gemcitabine, and cisplatin plus vinorelbine, have similar efficacy and different toxicity profiles, and they can treat advanced NSCLC [98]. The results from this study showed that doxorubicin, mitoxantrone, irinotecan may have inhibitory effects on *CD27* and *PD1* based on the CMap pharmacogenomic profiles [54], with potential implications in combined use with NSCLC immunotherapy.

Anisomycin, an antibiotic and a potential psychiatric drug [99,100], can activate stress-activated protein kinases, MAP kinase, and other signal transduction pathways [101]. Disulfiram (Antabuse), a drug for alcohol use disorders, is a possible treatment for cancer [102], parasitic infections [103], and latent HIV infection [104]. Puromycin is an antibiotic protein synthesis inhibitor critical in mRNA display [105]. Aztreonam is an antibiotic mainly used to treat infections. Piperlongumine is a natural alkaloid from long peppers, which increases the level of reactive oxygen species and selectively kills cancer cells [106,107]. Trichostatin A, an organic compound and an antifungal antibiotic, selectively inhibits class I and class II mammalian histone deacetylase (HDAC), but not class III HDAC [108]. Ivermectin, a medication used to treat many types of parasite infestations, showed in vitro antiviral effects against several positive-sense single-strand RNA viruses, including SARS-CoV-2 [109], with doses much higher than the maximum approved for safely use [110,111]. These results provided evidence for future drug repurposing study to improve NSCLC treatment in combination with clinical therapies.

## 4. Discussion

Lung cancer has the highest cancer-related mortality in both men and women due to its complex etiology, advanced disease stage at the time of diagnosis, and a lack of biomarkers for the selection of chemotherapy, immunotherapy, and radiotherapy in individual patients based on their tumor characteristics. Multiple ICIs have more CNV occurrences than mutations in NSCLC patient tumors. The mechanisms of CNV-mediated transcriptional perturbations in the NSCLC disease course are not well understood. This study developed the novel Boolean implication network methodology to model integrated CNV and transcriptomic profiles in NSCLC tumors. The constructed immune-omics networks contain novel NSCLC proliferation genes selected from the CRISPR-Cas9 and RNAi screening data and prognostic genes validated in extensive patient cohorts. These identified genes are gnomically/transcriptomically associated with ICIs, including *CD27*, *PD1*, and *PDL1*, and are linked to chemotherapy and radiotherapy response in NSCLC. The constructed immune-omics networks reveal important information underlying molecular mechanisms of NSCLC, with potential implications in developing novel therapeutic strategies and discovering repositioning drug candidates to improve clinical therapies.

NSCLC tumor immune microenvironment is a complex molecular network functioning in multi-dimensional components of immune and stromal cells as well as blood vessels, exerting intricate interactions with epithelial tumor cells. To model this complicated molecular machinery, the Boolean implication network scheme was developed to integrate genome-scale CNV and gene expression profiles in NSCLC tumors. The presented Boolean implication network algorithm has the following advantages over other existing models. Firstly, it can analyze discrete CNV data and continuous gene expression data in a seamless, biologically relevant way that correlation networks cannot. Secondly, unlike the acyclic Bayesian networks [112], the presented Boolean implication networks can model cyclic molecular interactions, such as feedback loops. Finally, the Boolean implication networks can model multivariate data based on prediction logic in combination with robust statistical tests, whereas other Boolean networks [113,114,115] are primarily focused on binary variables. This is an important improvement because categorized CNV and gene expression data should have at least three states to be biologically relevant, representing amplification/normal/deletion, or upregulation/normal/downregulation. We have utilized the Boolean implication networks to model genome-scale co-expression networks and crosstalk with major NSCLC signaling pathways [37,116,117]. Our Boolean implication networks identified sets of biomarkers that generated an accurate prediction of lung cancer risk and metastases [37,116,117]; meanwhile, Boolean implication networks revealed more biologically relevant molecular interactions than the Boolean networks [113,114,115], Bayesian networks, and Pearson’s correlation networks [118]. Based on the candidate genes identified with the Boolean implication networks from public microarray data, the NSCLC prognostic and chemo-predictive 7-gene qRT-PCR assay was developed using multiple patient cohorts collected from the US hospitals and was validated with a clinical trial JBR.10 [12]. Included in this 7-gene assay, CD27 and ZNF71 had concordant mRNA and protein expression in NSCLC tumors [12]. Our recent study showed that *ZNF71-KRAB*, the KRAB isoform that is transcriptional repression, was associated with EMT in NSCLC tumors and cell lines [119]. *CD27*, an emerging ICI [13], is being tested as adjuvant therapy in phase I/II clinical trials for multiple tumor types with promising results [14,15]. These results demonstrate that embedding molecular networks and crosstalk with major NSCLC signaling hallmarks can identify biologically relevant biomarkers with prognostic and predictive implications and with potential as therapeutic targets.

Novel combinations of chemotherapy with ICIs complicate biomarker discovery and multiple complex interactions relevant to immune responses should be considered in this process [120]. This study improved the Boolean implication networks to model multivariate CNV and gene expression profiles in NSCLC tumors and identified the immune-omics network of *CD27*, *PD1*, and *PDL1*. Among the identified five prognostic genes, anti-human FRα (FOSL1) antibody-drug conjugate (MORAb-202) achieved robust antitumor response in patient-derived xenograft models [83] and a phase I clinical trial of FRα (FOSL1)-positive advanced solid tumors [84]. The identified NSCLC proliferation genes (*COPA*, *CSE1L*, *EIF2B3*, *LSM3*, *MCM5*, *PMPCB*, *POLR1B*, *POLR2F*, *PSMC3*, *PSMD11*, *RPL32*, *RPS18*, and *SNRPE)* had a significant impact on 100% of NSCLC cell lines in CRISPR-Cas9 and RNAi assays. *TBP-1 (PSMC3)* depletion stabilizes *FRα (FOSL1)* and further increases its levels in tumor cells expressing RAS–ERK pathway oncogenes [121]. These results show that the proposed network approach reveals molecular interactions relevant to the underlying mechanisms. The Golgi apparatus, *M-COPA* (2-methylcoprophilinamide), might be a promising therapeutic target to conquer the detrimental cycle of TKI resistance in EGFR-mutated NSCLC cells via downregulating cell surface RTK expression [122]. *CSE1L* promotes the nuclear accumulation of transcriptional coactivator *TAZ* and enhances invasiveness and malignancy in human lung cancer and glioblastoma cells [123]. Depletion of core spliceosome encoding genes *SNRPE* or *SNRPD1* remarkably reduces cell viability in breast, lung, and melanoma cancer cell lines [124]. Knockdown of *SNRPE* leads to dramatically decreased mTOR mRNA and protein levels, accompanied with deregulated mTOR pathway [124]. Together, these results demonstrate that the multi-omics network approach unveils promising therapeutic strategies and novel linkages to ICIs including *CD27*, *PD1*, and *PDL1*. The association of the identified genes with chemoresponse and radiotherapy response revealed in this study can aid future clinical studies toward precision oncology.

It remains a challenge to cure early-stage NSCLC and improve survival outcomes due to the varying performance of lymphadenectomy, the limitations of current pathologic nodal staging, and our insufficient understanding of molecular features underlying tumor aggressiveness. Sentinel lymph node (SLN) mapping enables targeted nodal sampling for accurate staging of breast cancer and melanoma. Unfortunately, standard SLN mapping techniques with blue dye and radiocolloid tracers have not yielded reproducible results in lung cancer [125]. Recent intraoperative near-infrared (NIR) image-guided lung SLN mapping has emerged as a promising technology to identify lymph node micrometastases. The first reported analysis of NIR image-guided SLN mapping in NSCLC showed that patients with pN0 SLNs had favorable survival outcomes [126]. However, there were concerns that these results cannot be generalized to the overall NSCLC patients due to its small sample size and variable indocyanine green injection techniques for the NIR SLN mapping [127]. Inaccurate staging in early-stage NSCLC patients will hinder appropriate decisions of both intraoperative and adjuvant treatment. The development of clinically applicable prognostic biomarkers for early-stage NSCLC could potentially resolve this issue by identifying more aggressive tumors prone to recurrence/metastasis beyond the morphological assessment. Patients with molecularly more aggressive tumors might benefit from adjuvant therapy. The 5-gene prognostic model identified in this study has potential implications in clinics pending further validation. West Virginia University hospitals have used cancer patient sequencing data generated by Caris Life Sciences (Irving, TX) for clinical care. We are currently working with Caris Life Sciences to obtain these data for research as a network member. We also obtained an agreement to collaborate with Tempus (Chicago, IL) to utilize their de-identified database for research. The results from this study will be validated with the sequencing data of these external patient cohorts both retrospectively and prospectively in our next study.

Drug repositioning is an important pharmaceutical research and development (R & D) strategy to improve patient care, due to a lower risk of failure, expediated drug development, and less investment [22]. The presented network approach, in combination with comprehensive CRISPR-Cas9/RNAi screening and extensive patient gene expression/survival profiles, discovered promising drug repurposing candidates for treating NSCLC. Three chemotherapeutic agents were recommended for repurposing in this study: doxorubicin, mitoxantrone, and irinotecan. This study provided the first evidence that these agents had inhibitory effects on *CD27* and *PD1*, based on the CMap pharmacological database and the CCLE NSCLC panel. Cisplatin plus irinotecan was proven as effective in treating advanced NSCLC in a randomized phase III clinical trial [98], substantiating the validity of the presented approach. The identified antibiotic compounds (anisomycin, puromycin, aztreonam, trichostatin A, and ivermectin), natural compound Piperlongumine, and disulfiram (Antabuse) warrant further study as repositioning candidates for treatment of NSCLC, based on their relevant pharmacological profiles in the CMap database and their inhibitory effects on the CCLE NSCLC cell lines.

## 5. Conclusions

This study identified a CNV-mediated transcriptional regulatory network of *CD27*, *PD1*, and *PDL1* with substantiation in multiple NSCLC cohorts using Boolean implication networks. This immune-omics network contained prognostic genes validated in extensive patient cohorts and proliferation genes having a significant impact on 100% of NSCLC cell lines in CRISPR-Cas9 and RNAi assays. A 5-gene prognostic model was derived and validated for patient stratification in multiple cohorts, including TCGA data. Numerous identified genes were associated with chemoresponse and radiotherapy response in NSCLC cell lines and patient tumors. The constructed immune-omics network revealed promising therapeutic strategies and novel linkages to ICIs including *CD27*, *PD1*, and *PDL1.* This study provided the first evidence that doxorubicin, mitoxantrone, and irinotecan had inhibitory effects on *CD27* and *PD1*. Drug repurposing candidates were discovered for future pharmaceutical R & D strategies.

## 6. Patents

The 7-gene NSCLC prognostic and predictive assay is included in the US National Phase Patent Application No. 17/251,359 and International Non-Provisional Patent Application No. PCT/US20/23597. The results in this study are included in the US Provisional Patent Application No. 63234393.

## Figures and Tables

**Figure 1 cancers-13-04296-f001:**
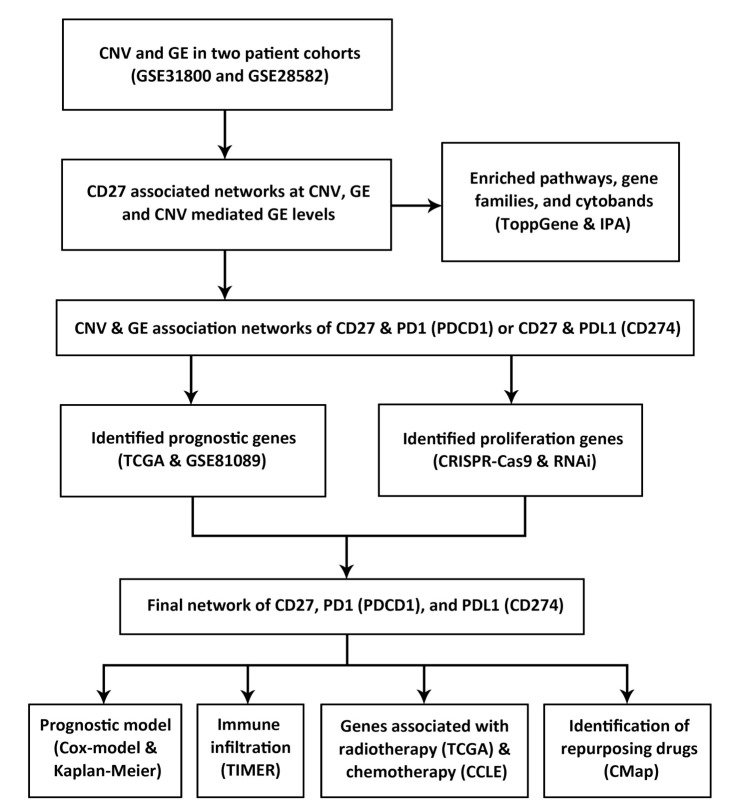
Overall study scheme. The arrows indicate analysis flow.

**Figure 2 cancers-13-04296-f002:**
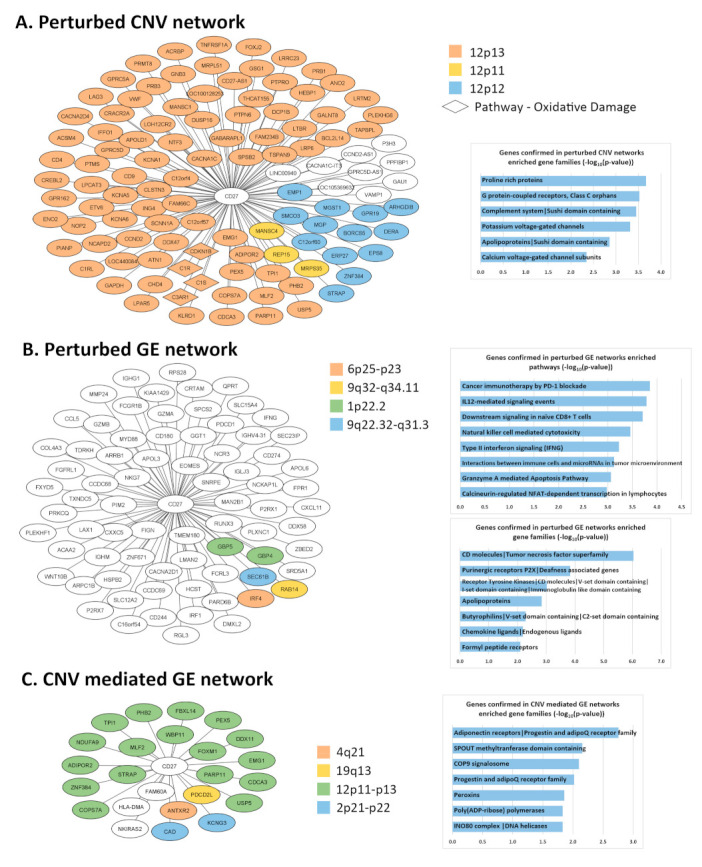
*CD27* multi-omics networks in NSCLC. (**A**) Perturbed *CD27* CNV network. The plot on the left shows the genes which had a significant (*p* < 0.05, *z-*tests) concordant co-occurrence of DNA copy number aberration with *CD27* in both patient cohorts GSE31800 and GSE28582. The bar chart on the right shows the –log_10_(*p*-values) of the significantly (*p* < 0.05, FDR < 0.05) enriched gene families in the ToppGene functional enrichment analysis. (**B**) Perturbed *CD27* gene expression (GE) network. The plot on the left shows the genes which had a significant (*p* < 0.05, *z-*tests) concordant co-expression with *CD27* in both patient cohorts GSE31800 and GSE28582. The bar charts on the right shows the –log_10_ (*p*-values) of the significantly (*p* < 0.05, FDR < 0.05) enriched pathways and gene families in the ToppGene analysis. (**C**) *CD27* CNV-mediated gene expression (GE) network. The plot on the left shows the genes which had a significant (*p* < 0.05, *z-*tests) concordant CNV-mediated gene expression regulation with *CD27* in both patient cohorts GSE31800 and GSE28582. The bar charts on the right shows the –log_10_ (*p*-values) of the significantly overrepresented gene families in the ToppGene analysis.

**Figure 3 cancers-13-04296-f003:**
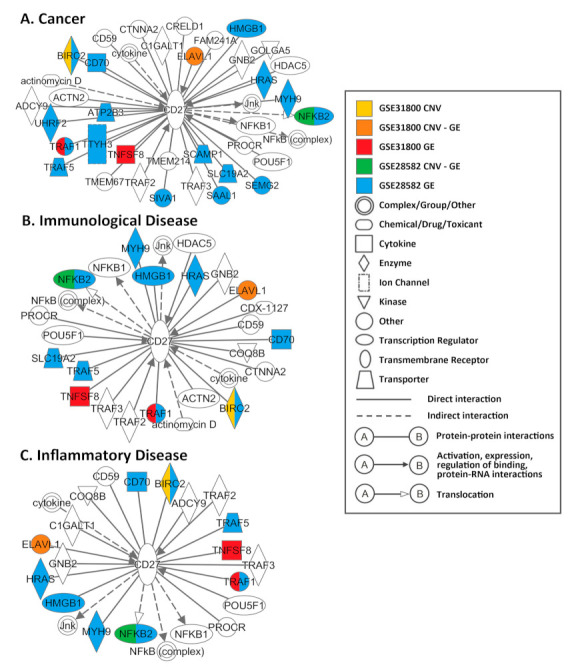
The experimentally confirmed interactions with *CD27* retrieved from IPA. Functional involvements in (**A**) cancer-related interactions, (**B**) immunological disease-related interactions, and (**C**) inflammatory disease-related interactions.

**Figure 4 cancers-13-04296-f004:**
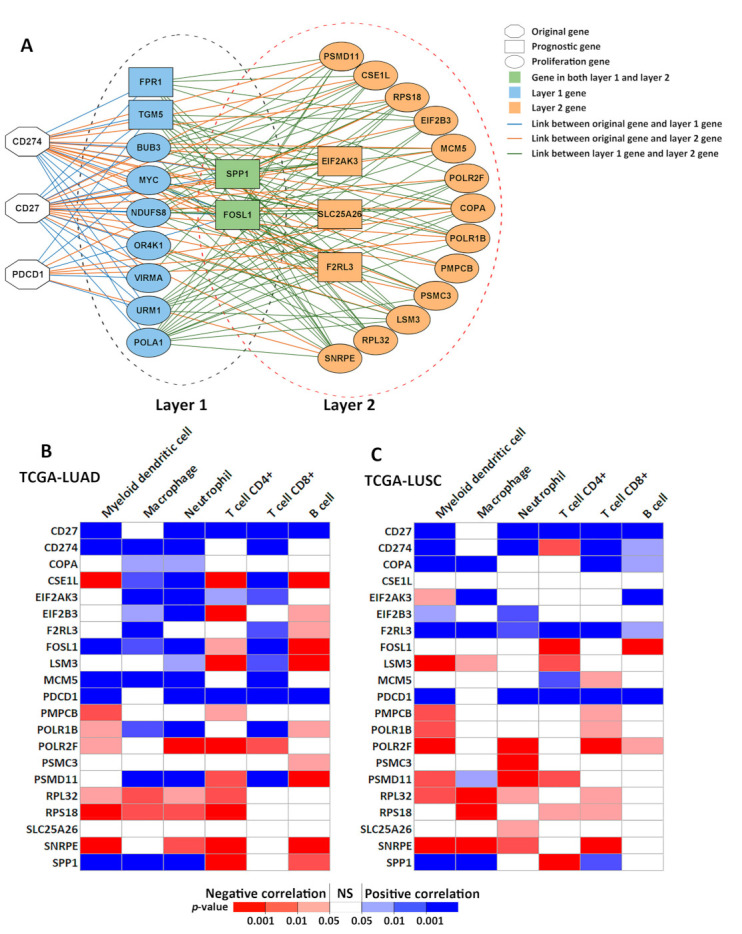
*CD27*, *PD1* and *PDL1 immune-omics* network in NSCLC. (**A**) *CD27*, *PD1*, and *PDL1* multi-omics association network. (**B**) The correlation of immune infiltration level with gene expression in TCGA-LUAD patients (*n* = 515) estimated with TIMER. (**C**) The correlation of immune infiltration level with gene expression in TCGA-LUSC patients (*n* = 501) estimated with TIMER. NS: not significant.

**Figure 5 cancers-13-04296-f005:**
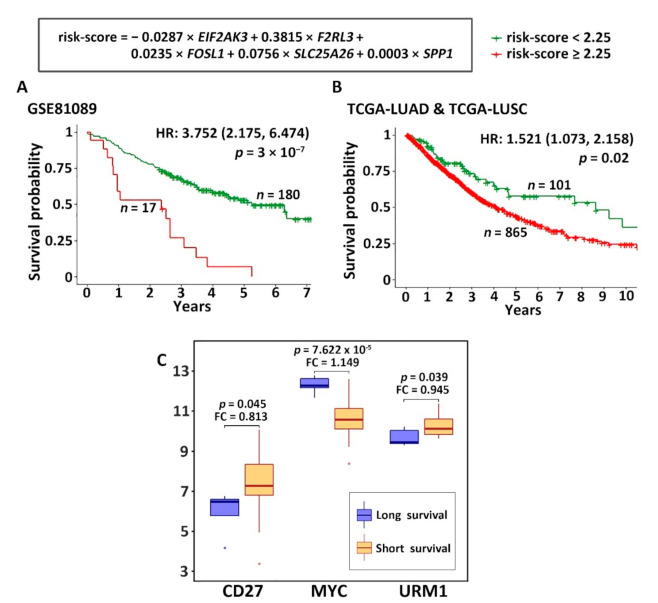
The 5-gene prognostic model for NSCLC. (**A**) Kaplan–Meier analysis of patients grouped by risk-scores with the cutoff value of 2.25 in GSE81089. (**B**) Kaplan–Meier analysis of patients stratified by risk-scores with the cutoff value of 2.25 in TCGA-LUSC and TCGA-LUAD. The plot shows the results of first 10 years after surgery. (**C**) Genes associated with radiotherapy response. These genes had a significant differential expression (*p* < 0.05, two-sample *t*-tests) in the long survival group versus the short survival group in TCGA-LUSC and TCGA-LUAD. Stage III or IV patients who had received radiotherapy were included in the analysis. FC: fold change of gene expression in the long survival versus the short survival patient group.

**Figure 6 cancers-13-04296-f006:**
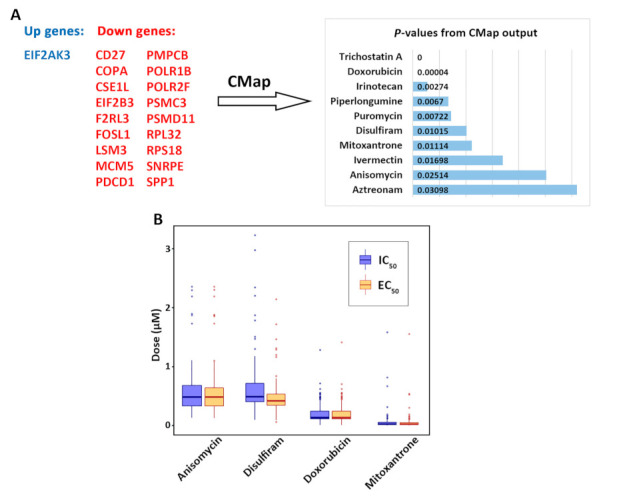
Discovery of repurposing drug candidates. (**A**) Selection of small molecules based on our identified gene expression signature with CMap. (**B**) Small molecules that had a low average concentration of drug response in the CCLE NSCLC cell lines (*n* = 117). An outlier, disulfiram EC_50_ = 25.645, was excluded from the plot.

**Figure 7 cancers-13-04296-f007:**
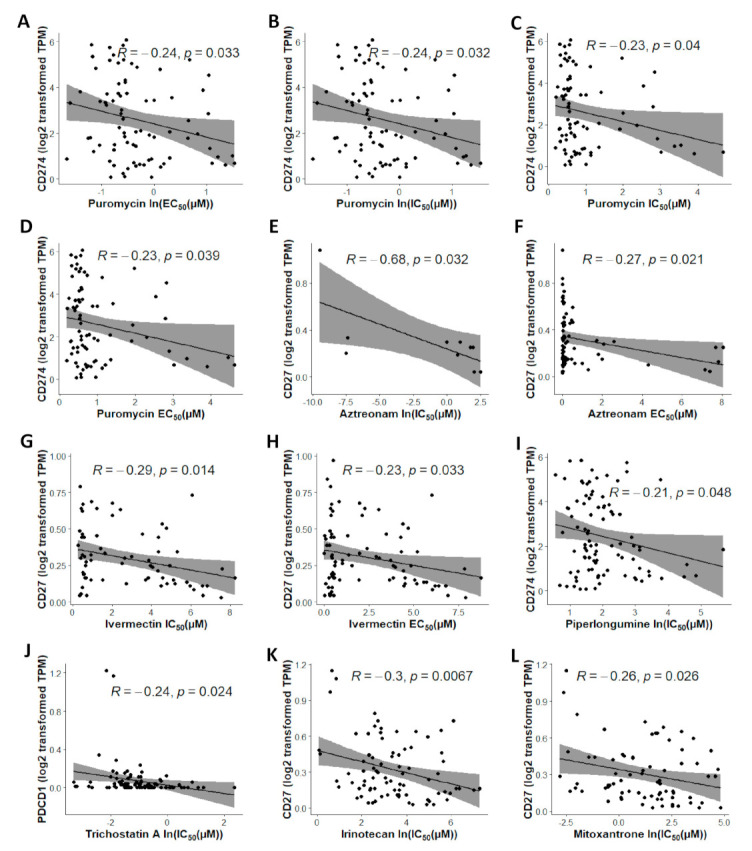
Compounds with a significantly negative correlation of drug concentration and the mRNA expression of *CD27*, *PD1*, or *PDL1* in the CCLE NSCLC cell lines (*n* = 117). The drug activity measurements included IC_50_, EC_50_, ln(IC_50_), and ln(EC_50_).

**Table 1 cancers-13-04296-t001:** Genes (shown in Figure 4A) with a significant differential expression (*p* < 0.05; two sample *t*-tests) in sensitive versus resistant CCLE NSCLC cell lines (*n* = 117) to specific drugs.

Variation	PRISM ln(IC_50_)	PRISM ln(EC_50_)	GDSC1 ln(IC_50_)	GDSC2 ln(IC_50_)
Carboplatin				
Cisplatin				*PSMD11*
Docetaxel		*RPS18*	*EIF2AK3*, *PMPCB*, *PSMD11*	*PMPCB*, * POLR1B*
Erlotinib		*LSM3*, *POLR2F*,* RPL32*	*CSE1L*	*FOSL1*
Etoposide		*PSMD11*		
Gefitinib			*COPA*	
Gemcitabine	*PSMC3*	*PSMC3*		*SPP1*, * MCM5*, *RPS18*
Paclitaxel	*PSMD11*	*PMPCB*, * RPS18*		*POLR1B*
Pemetrexed			*RPS18*	
Vinorelbine		*POLR2F*	*CSE1L*, * SNRPE*	

Drug activity measurements were natural log transformed. Red font indicates the gene has a higher expression in the resistant cell lines than in the sensitive cell lines. Blue font indicates the gene had a lower expression in the resistant cell lines than in the sensitive cell lines.

## Data Availability

Access to data available from public domains is provided in the manuscript.

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
