# Peer review of "Immune-Omics Networks of CD27, PD1, and PDL1 in Non-Small Cell Lung Cancer"

_cancers, 2021, doi:10.3390/cancers13174296_

Round 1

Reviewer 1 Report

The predicted gene signature is not supported by experimental evidence. These findings are suitable for more specially journals on bioinformatic prediction. 

Author Response

The 5-gene prognostic model was identified from RNA-sequencing data of an NSCLC patient cohort (n = 197) and was validated with TCGA NSCLC patients (n = 1,163). The identified immune-omics networks of CD27, PD1, and PDL1 were also validated in multiple patient cohorts (n = 371). All the model details and access to the publicly available patient data are provided in the manuscript to ensure the reproducibility of the results. Leveraging the extensive public data consortia has greatly expediated the scientific discovery. These results have potential prognostic implications pending further clinical validation. West Virginia University hospitals have used cancer patient sequencing data generated by Caris Life Sciences (Irving, TX) for clinical care. We are currently working with Caris Life Sciences to obtain these data for research with the existing data use agreement. We also obtained an agreement to collaborate with Tempus (Chicago, IL) to utilize their de-identified database for research. The results from this study will be validated with the sequencing data of these external patient cohorts both retrospectively and prospectively in our next study.

Reviewer 2 Report

Thank you for giving me the opportunity to review your work entitled "Immune-omics networks of CD27, PD1, and PDL1 in non-small cell lung cancer". I congratulate the authors for the excellent work they have done and the amount of work this has required. I believe that in the future the keystone in the treatment of lung neoplasms lies above all with the greater knowledge of these mechanisms and of what lies at the basis of the immune response. For this reason I find your work very innovative in this sense, the applied methodology is rigorous and well done, the results are clear and well explained. In light of this, I have no particular comments or criticisms to make, perhaps the translation from the laboratory result to the clinical prognostic stratification is yet to be verified with further studies. However, overall great job congratulations

Author Response

We thank the reviewer for the very positive comments. Indeed, West Virginia University hospitals have used cancer patient sequencing data generated by Caris Life Sciences (Irving, TX) for clinical care. We are currently working with Caris Life Sciences to obtain these data for research with the existing data use agreement. We also obtained an agreement to collaborate with Tempus (Chicago, IL) to utilize their de-identified database for research. The results from this study will be validated with the sequencing data of these external patient cohorts both retrospectively and prospectively in our next study.

Reviewer 3 Report

Excellent study ! I would congratulate with authors for this interesting and well written paper . 

Do you think that this study is applicable to a SLN mapping in early staging NSCLC?

We are now in a new era in cancer therapy therefore the knowledge of dependable

 biomarkers for the selection of patients who can benefit from immunotherapeutic or targeted interventions is an essential part of the oncological pathway.

Given that I find this study original which has been conducted with a rigorous methodology with clear and promising results. I, therefore, believe that taking into account the innovation of the work and its originality, I would like to suggest its clinical application in the stratification of early cancer patients for the clinical verification of these preliminary data. In fact, I believe that this could be a great contribution also through further studies in this specific field.

That said I have no additional comments or suggestions except my congratulations on the excellent work

Author Response

We appreciate the very positive review and the excellent comments on SLN mapping. The following discussion is now added to the manuscript (lines 661-685).

It remains a challenge to cure early-stage NSCLC and improve survival outcomes due to the varying performance of lymphadenectomy, the limitations of current pathologic nodal staging, and our insufficient understanding of molecular features underlying tumor aggressiveness. Sentinel lymph node (SLN) mapping enables targeted nodal sampling for accurate staging of breast cancer and melanoma. Unfortunately, standard SLN mapping techniques with blue dye and radiocolloid tracers have not yielded reproducible results in lung cancer [126]. Recent intraoperative near-infrared (NIR) image-guided lung SLN mapping has emerged as a promising technology to identify lymph node micrometastases. The first reported analysis of NIR image-guided SLN mapping in NSCLC showed that patients with pN0 SLNs had favorable survival outcomes [127]. However, there were concerns that these results cannot be generalized to the overall NSCLC patients due to its small sample size and variable indocyanine green injection techniques for the NIR SLN mapping [128]. Inaccurate staging in early-stage NSCLC patients will hinder appropriate decisions of both intraoperative and adjuvant treatment. The development of clinically applicable prognostic biomarkers for early-stage NSCLC could potentially resolve this issue by identifying more aggressive tumors prone to recurrence/metastasis beyond the morphological assessment. Patients with molecularly more aggressive tumors might benefit from adjuvant therapy. The 5-gene prognostic model identified in this study has potential implications in clinics pending further validation. West Virginia University hospitals have used cancer patient sequencing data generated by Caris Life Sciences (Irving, TX) for clinical care. We are currently working with Caris Life Sciences to obtain these data for research as a network member. We also obtained an agreement to collaborate with Tempus (Chicago, IL) to utilize their de-identified database for research. The results from this study will be validated with the sequencing data of these external patient cohorts both retrospectively and prospectively in our next study.